# Hemolytic Activity and Cytotoxicity of Synthetic Nanoclays with Montmorillonite Structure for Medical Applications

**DOI:** 10.3390/nano13091470

**Published:** 2023-04-25

**Authors:** Olga Yu. Golubeva, Yulia A. Alikina, Elena Yu. Brazovskaya, Nadezhda M. Vasilenko

**Affiliations:** Laboratory of Silicate Sorbents Chemistry, Institute of Silicate Chemistry of Russian Academy of Sciences, Adm. Makarova emb., 2, 199034 St. Petersburg, Russia

**Keywords:** montmorillonite, nanoclays, nanocarriers, drug delivery, hemosorption, enterosorption, hemolytic activity, cytotoxicity, MTT assay

## Abstract

The factors influencing the appearance of toxicity in samples of synthetic montmorillonite with a systematically changing chemical composition Na_x_(Al, Mg)_2-3_Si_4_O_10_(OH)_2_ nH_2_O, which are potentially important for their use in medicine as drug carriers, targeted drug delivery systems, entero- and hemosorbents have been studied. Samples synthesized under hydrothermal conditions had the morphology of nanolayers self-organized into the nanosponge structures. The effect of the aluminum content, particle sizes, porosity, and ζ-potential of the samples on their toxicity was studied. The cytotoxic effect of the samples on eukaryotic cells *Ea. hy 926* was determined using the MTT assay. The hemolytic activity of the samples in the wide concentration range in relation to human erythrocytes was also estimated. It has been established that the toxicity of aluminosilicate nanoparticles can be significantly reduced by correctly selecting their synthesis conditions and chemical composition, which opens up the opportunities for their use in medicine.

## 1. Introduction

At the moment, there are several promising areas of application of porous aluminosilicate nanostructures in medicine, which are at the stages of laboratory research or clinical trials. These areas include the development of sustained release drug carriers, targeted drug delivery systems, nanovaccines, entero-, hemo-, and application sorption, materials for purification of fibrinogen and lipoproteins [1,2,3,4,5,6,7,8,9,10,11,12].

The porous structure and developed inner surface of montmorillonites determines their unique set of properties, which combines high sorption capacity, high specific surface area, ion exchange capacity, a wide range of active centers on the surface, and biocompatibility [6,13,14]. These characteristics make montmorillonite (MT) very promising for its use in medicine. In addition, among all porous aluminosilicates and clay minerals, MT have the greatest adsorption ability for inorganic and organic cations, proteins, as well as drugs [15,16,17]. The unique adsorption characteristics of montmorillonite are due to the peculiarities of its structure, namely, the ease of increasing the interlayer distance due to adsorption and intercalation of various molecules within a very wide range, up to complete division into separate layers [18,19,20,21,22]. This property of MT is widely used to obtain various adsorption materials, in particular medical appointment, such as enterosorbents. The use of MT in medicine allows to solve the problems of removing allergens, toxins, viruses, inflammatory mediators from the body, as well as preventing their movements into the systematic bloodstream [23,24,25,26].

The development of devices and methods that ensure the extraction of pathogenetically significant compounds from the blood is extremely relevant. The problem of sepsis and septic shock is acute before modern medicine. In the recent past, sepsis ended in death in 80% of cases [27]. Recent studies have shown that it is possible to reduce mortality several times using modern extracorporeal detoxification methods-hardware cleansing of blood outside the body using physico-chemical methods, in particular using sorption methods [28]. In this regard, the search for effective and safe sorbents for extracorporeal detoxification is an urgent and socially significant task.

Currently, the main developments in the field of selective hemosorbents are carried out using synthetic and natural polymers. Carbon hemosorbents are widely used, however, they do not have selectivity and damage blood cells. High adsorption capacity of montmorillonites makes it possible to consider them as a potential material for medicine, in particular, for hemosorption. In this case, the study of cytotoxicity and hemolytic activity in samples is of great importance.

Hemolysis is the destruction of red blood cells in a blood sample, with the release of various biologically active substances from them and, most importantly, hemoglobin into plasma. The study of hemolysis of samples can be considered as a model study for their effect on biological membranes, since usually hemolysis correlates with other tests for cytotoxicity [29,30,31].

The toxicity of raw aluminosilicate minerals significantly limits the possibilities of their application in medicine. Thus, studies of hemolysis in vitro have shown the presence of hemolytic activity for a number of silicate minerals: attapulgite, chrysotile, sepiolite, palygorskite, kaolinite, montmorillonite, and illite [32,33,34,35]. It was shown [35] that heat treatment at 600 °C can reduce the hemolytic activity of the minerals. Several assumptions have been made about the presence of hemolytic activity (HA) in silicate minerals, in particular, a number of researchers have found correlations between HA and mineral surface charge [36], chemical compositions [37], particle sizes, and specific surface area [38].

Cytotoxicity studies have shown the presence of toxicity in raw clay minerals. In most cases, the harmful effect is associated with the presence of impurity phases. Thus, the cytotoxicity of natural kaolinite is most likely associated with the presence of quartz [39]. Raw MT studies have shown that it can cause cytotoxic effects at high concentrations after long-time exposure [40].

The reasons for the presence of toxicity are not completely clear. If we talk about the toxicity of inorganic nanoparticles in general, then the following parameters that affect toxicity are distinguished: particle size and specific surface area, particle shape, shape factor, degree of crystallinity, surface properties, and susceptibility to agglomeration.

Directed synthesis makes it possible to solve the problem of the variability of the chemical and phase composition of raw minerals, which are the main cause of toxicity. In addition, directed hydrothermal synthesis makes it possible to obtain silicates with specified characteristics-specific chemical composition, given particle sizes and morphology, and certain surface properties. 

In this work, studies of the hemolytic activity and cytotoxicity of samples of synthetic porous aluminosilicates with the montmorillonite structure, which are characterized by certain porous-textural, microstructural characteristics, specified by the chemical composition and particle size, were carried out for the first time. Such particles can be considered as model objects to identify possible patterns between the physicochemical characteristics of inorganic nanoparticles and their toxicity.

## 2. Materials and Methods

### 2.1. Synthesis

Synthesis was carried out by hydrothermal treatment of dried gels of appropriate compositions in steel autoclaves with platinum crucibles. The composition of the initial gels was calculated based on the ideal formula of montmorillonite, which has the following form Na_x_(Al, Mg)_2-3_Si_4_O_10_(OH)_2_·*n*H_2_O. The surface charge deficit *x* was varied from 0 to 1.9 by changing the aluminum content in the composition of the initial gel, assuming that part of the magnesium in the octahedral layers is isomorphically substituted by aluminum. Starting gels were prepared using tetraethoxysilane TEOC ((C_2_H_5_O)_4_Si, special purity grade, ≥99.0%, Sigma, Stainheim, Germany), magnesium nitrate Mg(NO_3_)_2_·6H_2_O (reagent grade, Vecton, St.Petersburg, Russia), aluminum nitrate Al(NO_3_)_3_·9H_2_O (reagent grade, ≥97.0%, Vecton, St.Petersburg, Russia), HNO_3_ (reagent grade, 65 wt %, NevaReactiv, St.-Peterburg, Russia), ammonia NH_4_OH (special purity grade, NevaReactiv, St.-Peterburg, Russia), and ethanol C_2_H_5_OH (96 wt %). Hydrothermal treatment of gels was carried out at 350 °C (in some cases-at 250 °C) for 2–4 days at an autogenous pressure of 70 MPa.

### 2.2. Characterization

The samples were characterized by the complex of the physicochemical analysis methods. Chemical analysis was carried out for the content of silicon, magnesium, aluminum, and sodium. The samples were studied by X-ray phase analysis; the analysis of porous-textural characteristics was carried out using the methods of low-temperature nitrogen adsorption and benzene adsorption; the morphology of the samples was studied using electron microscopy methods; the surface properties were evaluated by analyzing the ζ-potential of the samples; the cation exchange capacity (CEC) was determined. In addition, studies of hemolysis by synthesized samples were carried out, as well as an assessment of their cytotoxicity against of human endothelial cells *Ea. hy 926* using the MTT test [41]. Human endothelial cells *Ea.hy 926* were derived from the American Type Culture Collection (ATCC, cat. CRL 2922, Manassas, VA, USA). Measurements were carried out at least three times for each test sample. The results are shown as a mean ± standard deviation. The methods used are described in more detail in Appendix A.

## 3. Results and Discussion

The results of the chemical analysis of the samples (Table 1) confirmed that samples with different aluminum content were obtained.

Studies of the samples by X-ray phase analysis confirmed the obtaining of single-phase samples with the montmorillonite structure with an average particle size of 20 ± 3 nm. X-ray patterns of the samples and their analysis are presented in the Appendix A.

According to the SEM analysis, all samples are characterized by a layered morphology typical of montmorillonites (Figure 1). Thin nanolayers self-organize into larger aggregates with the formation of secondary porous structures, such as nanosponges, which is clearly seen in Figure 1c, which shows the results of the study of the samples using the FIB-SEM method.

The results of ζ-potential determination (Figure 2) showed that all samples have a negative surface charge, which is typical for aluminosilicates. The zeta potential becomes more negative when the degree of substitution of magnesium for aluminum increases. At the same time, the dependence of the cation exchange capacity (CEC) of the samples on the composition has a different character-it increases with the increase in the content of aluminum oxide in the samples and passes through an extremum in the region of 50% substitution of magnesium for aluminum. The same trend, similar to the change in CEC of MT samples with their composition, was also found on the dependences of the degree of adsorption of thiamine on MT samples [42].

The specific surface area determined from the data of low-temperature nitrogen adsorption (Table 2) decreases with increasing aluminum content in the samples under study and varies from 100 up to 320 m^2^/g to 106 m^2^/g.

As can be seen from the obtained results of the influence of the composition of the samples on their hemolytic activity (HA), presented in Figure 3, all studied samples are toxic at a concentration of 10 mg/mL. The dependence of HA on the aluminum content in the samples is visible. As the aluminum content increases from 0 to 22–22%, there is a significant increase in HA (% hemolysis) up to 87%, which means a significant proportion of destroyed erythrocytes. It should be noted that for a sample that does not contain aluminum, Al0, HA is practically absent. Thus, a negative effect on the appearance of toxicity is revealed, which is exerted by the aluminum content in the composition of the samples. At the same time, the nature of the change in HA—an increase up to the composition of Al1.0 followed by a slight decrease, fully correlates with the dependence of the cation exchange capacity of the samples (Figure 3), and also there is a correlation between the measurement of HA and the course of changes in the volume of mesopores, determined from the data of benzene adsorption.

Figure 4 presents the results of studying the HA of the MT samples of various chemical compositions depending on the concentration of the sample. It follows from the data obtained that the HA of the samples increases with increasing sample concentration, however, the nature of growth for samples of different compositions is different. For example, samples of Al0 and Al0.2 compositions can be considered as non-toxic up to concentrations of 2.5 mg/g. If the hemolytic activity is less than 5%, the material will have no toxic effect and conform to the requirements of the hemolysis test for medical materials [43]. With a subsequent increase in the sample concentration, a sharp increase in hemolysis is observed in the Al0.2 sample, while the Al0 sample can be considered non-toxic in the entire range of the studied concentrations, since the hemolysis of erythrocytes in its presence does not exceed 5% even at a sample concentration of 10 mg/mL. The Al1.0 sample exhibits significant HA at any studied concentrations.

The results of the cytotoxicity study of MT samples of different chemical composition correlate with the results of the study of their HA and demonstrate the relationship between the toxicity of samples and their chemical composition. Therefore, as can be seen from Figure 5, the highest toxicity, accompanied by the lowest cell survival in the entire range of studied samples, is characteristic of aluminum-containing samples. An increase in the aluminum content in the composition of the samples is accompanied by an increase in cytotoxicity. The Al0 sample is characterized by the least cytotoxicity.

As shown earlier, the synthesis conditions affect the properties and morphology of MTs. Under hydrothermal conditions for 1–3 days, the samples obtained are thin nanolayers organized into a nanosponge structure. An increase in the duration of synthesis to 3–10 days or more leads to the growth of crystals along the c axis and to the formation of a package structure, while the particle size in the direction perpendicular to axis c both according to TEM data, and according to the results of calculations by the Scherrer formula shows little change. The formation of a packet structure is accompanied by a decrease in the specific surface area of the samples.

Figure 6 shows the effect of synthesis conditions on the hemolytic activity of samples of two compositions, Al0.2 and Al1.0, obtained at different synthesis times. As can be seen from the presented results, the formation of the MT packet structure and the growth of crystals along the *c* axis is accompanied by a significant increase in their hemolytic activity, and, consequently, toxicity. This explains the presence of significant toxicity in natural clay minerals. Directed synthesis conditions make it possible to control the growth of crystals and obtain them in the form of nanolayers rather than plates, which can significantly reduce their toxicity.

It was shown [35] that heat treatment at 600 °C can reduce the hemolytic activity of a number of minerals. The results of the present study did not reveal such a trend for synthetic MT samples. Samples of MT of different chemical composition were subjected to additional heat treatment for 2 h at 550 °C. The results of the HA study of the samples before and after treatment (Table 3) show that HA not only does not decrease after heat treatment, but in most cases tends to increase. The only exception is the sample with the highest aluminum content, for which a slight decrease in HA is observed after heat treatment.

It is known that the adsorption properties of MT and the numerical values of its active adsorption sites are very sensitive to temperature changes. At the same time, when heated, only the water content in the clay changes and no structural changes occur in the range up to 800–900 °C. Therefore, the observed increase in HA can be attributed precisely to the removal of physical and structural water, which may have led to a greater availability of active sites on the MT surface and an increase in toxicity. Indirectly, this trend is also confirmed by the previously described correlation between the growth of HA and the growth of mesopores in the structure of the samples.

## 4. Conclusions

Studies of the hemolytic activity and cytotoxicity of synthetic nanoclays with montmorillonite structure showed a certain level of toxicity in all samples. At the same time, the results obtained make it possible to identify the main patterns and factors influencing the appearance of toxicity in the studied samples, as well as to determine the optimal conditions for the synthesis of montmorillonite, which make it possible to obtain samples with minimal toxicity.

Thus, it was found that the content of aluminum in the composition of the samples dramatically affects the appearance of toxicity. As the aluminum content increases from 0 to 20–22 wt %, there is a significant increase in HA (% hemolysis) up to 87%, which means a significant proportion of destroyed erythrocytes. It should be noted that for a sample that does not contain aluminum, HA is practically absent. Aluminum free samples can be considered non-toxic in a wide range of concentrations (0.1–10 mg/mL) and suitable for their use in medicine. However, the conditions for sample synthesis should be taken into account. As studies have shown, an increase in the duration of sample synthesis from 1–3 days to 12 days is accompanied by the disappearance of the nanosponge structure and the appearance of a packet structure, which leads to a sharp increase in both cytotoxicity and hemolytic activity. In addition, a correlation was found between the porous-textural characteristics of the samples and the presence of toxicity in them—an increase in the volume of mesopores in the structure of the samples is accompanied by an increase in hemolytic activity. At the same time, no dependence of hemolytic activity on the zeta potential of the samples was found.

## Figures and Tables

**Figure 1 nanomaterials-13-01470-f001:**
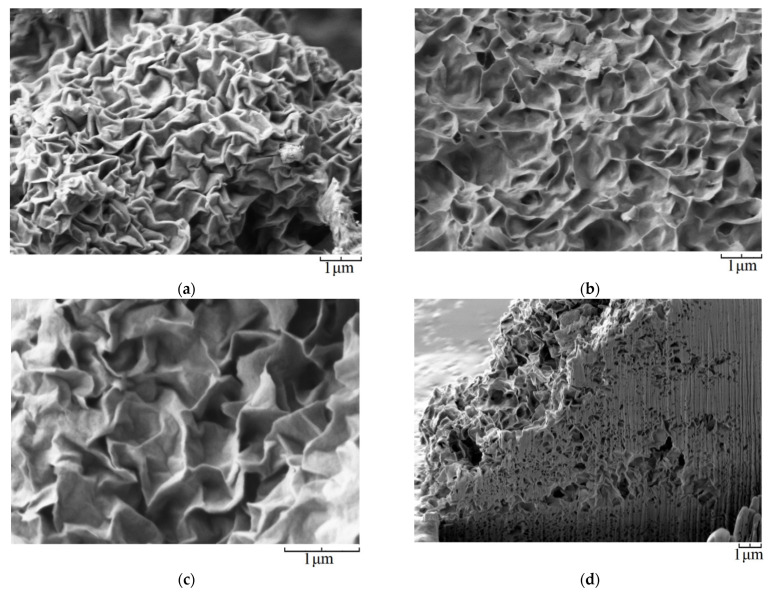
The results of the study of samples by scanning electron microscopy: (**a**)—Al0, (**b**)—Al0.2, (**c**)—Al1.0, (**d**)—Al1.8 (FIB-SEM image).

**Figure 2 nanomaterials-13-01470-f002:**
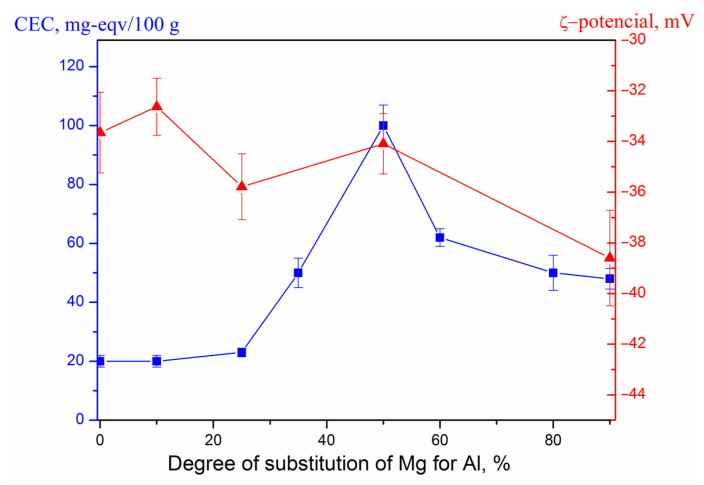
Dependence of the cation exchange capacity (■) and zeta potential (▲) of MT samples on the degree of substitution of magnesium for aluminum in octahedral layers.

**Figure 3 nanomaterials-13-01470-f003:**
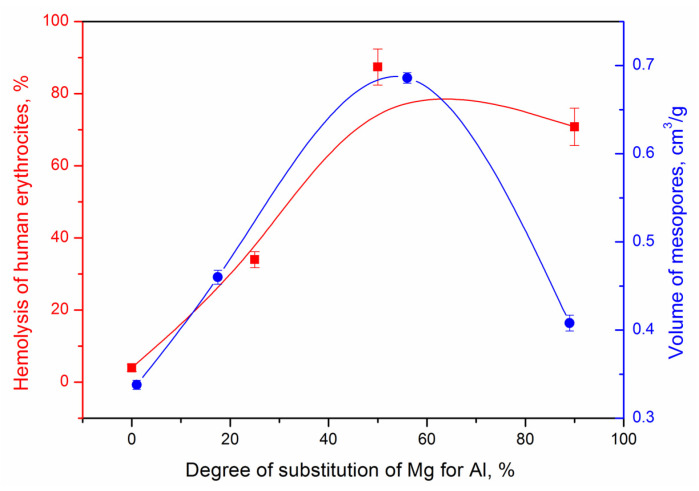
Dependence of hemolytic activity and mesopore volume in MT samples on the degree of substitution of magnesium for aluminum in them.

**Figure 4 nanomaterials-13-01470-f004:**
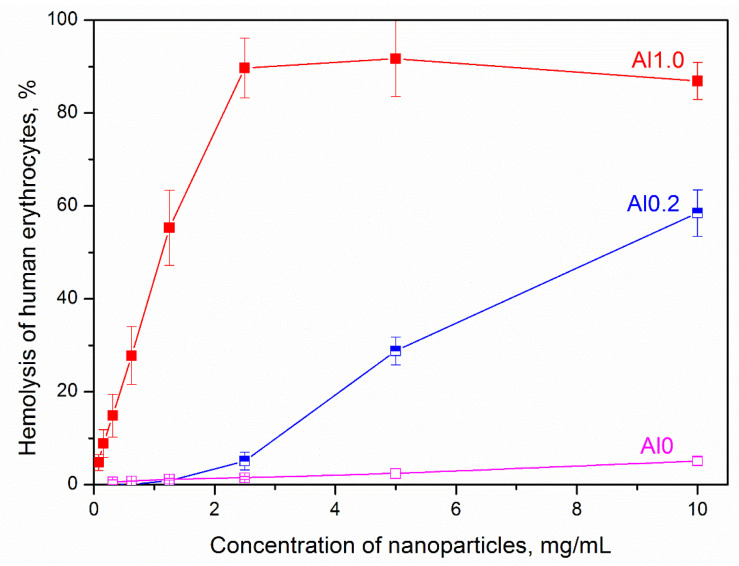
Hemolytic activity of the samples. Sample designations are given in accordance with Table 1.

**Figure 5 nanomaterials-13-01470-f005:**
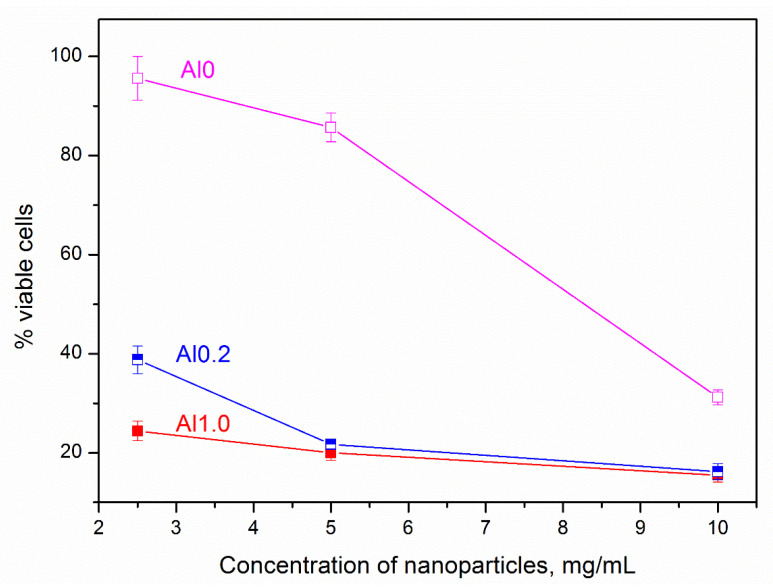
Dependence of cell survival on samples concentration.

**Figure 6 nanomaterials-13-01470-f006:**
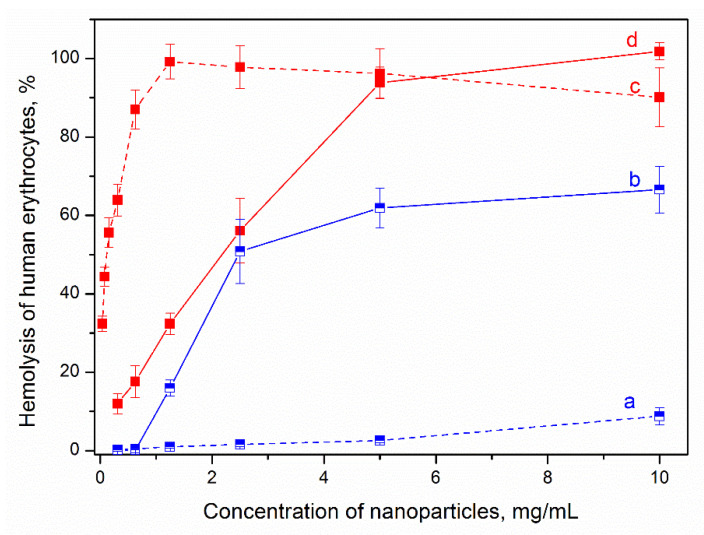
Effect of sample synthesis time on their hemolytic activity: (a)—Al0.2, 2 days; (b)—Al0.2, 12 days; (c)—Al1.0, 1 day; (d)—Al1.0, 12 days.

**Table 1 nanomaterials-13-01470-t001:** Chemical composition and denotation of some samples studied.

Sample Denotation	Composition by Synthesis	Oxide Contents by Analysis, wt %			
		SiO_2_	Al_2_O_3_	MgO	Na_2_O
Al0	Na_x_Mg_3_Si_4_O_10_(OH)_2_·nH_2_O	54.11	-	32.52	0.11
Al0.2	Na_x_Al_0.2_Mg_1.8_Si_4_O_10_(OH)_2_·nH_2_O	58.10	5.32	18.31	3.52
Al0.5	Na_x_Al_0.5_Mg_1.5_Si_4_O_10_(OH)_2_·nH_2_O	56.01	12.08	13.73	3.47
Al1.0	Na_x_Al_1.0_Mg_1.0_Si_4_O_10_(OH)_2_·nH_2_O	53.89	22.82	8.04	3.2
Al1.8	Na_x_Al_1.8_Mg_0.2_Si_4_O_10_(OH)_2_·nH_2_O	56.96	24.81	2.10	2.99

**Table 2 nanomaterials-13-01470-t002:** Parameters of the porous structure of MT samples determined from the data of benzene adsorption (W_s_, V_mic_, V_mes_) and low-temperature nitrogen adsorption (SSA).

Samples Denotation	SSA, m^2^/g	W_s_, cm^3^/g	V_mic_, cm^3^/g	V_mes_, cm^3^/g
Al0.2	320	0.509	0.171	0.338
Al0.5	207	0.569	0.109	0.460
Al1.0	190	0.722	0.036	0.686
Al1.8	106	0.474	0.066	0.408

SSA—specific surface area, BET method; W_s_—limiting volume of sorption space; V_mic_—micropore volume; V_mes_—mesopores volume.

**Table 3 nanomaterials-13-01470-t003:** Hemolytic activity of samples (10 mg/mL) before and after heat treatment at 500 °C.

Sample Denotation	% Hemolysis before Heat Treatment	% Hemolysis after Heat Treatment
Al0	4.2 ± 0.3	19.5 ± 0.6
Al0.5	34.3 ± 1.0	54.6 ± 1.2
Al1.0	86.9 ± 1.2	69.1 ± 3.4
Al1.8	71.2 ± 2.1	54.0 ± 0.7

## Data Availability

Data available on request due to restrictions eg privacy or ethical.

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
