# Peer review of "Hemolytic Activity and Cytotoxicity of Synthetic Nanoclays with Montmorillonite Structure for Medical Applications"

_nanomaterials, 2023, doi:10.3390/nano13091470_

Round 1

Reviewer 1 Report

This manuscript reported the hemolytic activity and cytotoxicity of synthetic nanoclays with montmorillonite structure from the aluminum content, particle sizes, porosity and Z-potential aspects. The hemolytic activity and cytotoxicity of samples of synthetic porous aluminosilicates with the montmorillonite structure made by directed hydrothermal synthesis sounds somehow interesting. In my opinion, this work  can be published in this journal after the following major questions are addressed:

(1) Line 42: "as well preventing their movements..." should be "as well as preventing their movements..."

(2) In Figure 2, the y-axis was written as COC, should it be CEC?

(3) Line 160: "As the aluminum content increases from 0 to 22-22 %", Actually from Figure 3, the aluminum content increased from 0 to 50-58 %. please check it and make correct discussion.

(4) The sample numbers listed in Table 1 and Table 2 are not consistent, Sample number Al1.0 should be added instead of Al1.2 in Table 2, because the following discussion is all related about the Al1.0.

(5)Line 244: "As the aluminum content increases from 0 to 22- 22 wt %" should be corrected.

(6) How about the size of the nanoparticles used in Figure 6, the data should be given according to the measurement. The investigation of size effect on HA should be carried out.

(7) The effect of Mg and Na contents on HA should be discussed besides Al. 

(6) The morphology effect (crystal or nanosponge) on HA should also be considered.

(7) References' format should be revised, shuch as the last page and the DOI of the references should be indicated consistently.

Author Response

Response to Reviewer 1 Comments

The authors would like to thank the Reviewer for a thorough reading of the manuscript and for valuable comments, which the authors took into account and made the necessary changes to the text of the manuscript.

Point 1. Line 42: "as well preventing their movements..." should be "as well as preventing their movements..."

Response 1. Thank you very much for this comment. Appropriate changes have been made.

Point 2. In Figure 2, the y-axis was written as COC, should it be CEC?

Response 2. Thank you very much for this remark. Indeed, there must be CEC. Appropriate changes have been made.

Point 3. Line 160: "As the aluminum content increases from 0 to 22-22 %", Actually from Figure 3, the aluminum content increased from 0 to 50-58 %. please check it and make correct discussion.

Response 3. The fact is that the X axis of Figure 3 shows not the content of aluminum oxide, but the degree of substitution of magnesium for aluminum. A sample with a degree of substitution of about 58 % (composition Al1.0) corresponds to an aluminum oxide content of 22 % (table 1).

Point 4. The sample numbers listed in Table 1 and Table 2 are not consistent, sample number Al1.0 should be added instead of Al1.2 in Table 2, because the following discussion is all related about the Al1.0.

Response 4. Thank you very much for this comment. Indeed, an error was made in the designations of the samples in Table 2. Appropriate corrections have been made.

Point 5. Line 244: "As the aluminum content increases from 0 to 22- 22 wt %" should be corrected.

Response 5. Thank you once again for your comments. Appropriate corrections have been made.

 Point 6. How about the size of the nanoparticles used in Figure 6, the data should be given according to the measurement. The investigation of size effect on HA should be carried out.

Response 6. Thе information about particles sizes is given in the text of manuscript on lines 124-126. An analysis of the X-ray diffraction data and calculations using the Scherrer formula showed that the particle size of all samples is approximately the same and is 20 ± 3 nm. X-ray patterns of the samples and their analysis are presented in the Supplementary information file (S4, Fig. S1).

Point 7. The effect of Mg and Na contents on HA should be discussed besides Al.

Response 7. As follows from the chemical analysis data (Table 1), the sodium content in the studied samples (except for the Al0 sample) is almost the same and is at the level of 3.2±0.2%. The magnesium content in the samples changes inversely proportional to the aluminum content, since part of the magnesium in the structure of the samples is replaced by aluminum. Whereas aluminum is a toxic element, it is more logical to associate the increase in hemolytic activity with an increase in the aluminum content, and not with a decrease in the magnesium content.

 Point 8. The morphology effect (crystal or nanosponge) on HA should also be considered.

Response 8. According to the SEM analysis, all samples are characterized by a layered morphology typical of montmorillonites. Thin nanolayers self-organize into larger aggregates with the formation of secondary porous structures, such as nanosponges. The formation of nanospongy secondary structures is characteristic of all samples. Therefore, the influence of morphology on the results of the study of the biological activity of samples can be neglected.

Point 9. References' format should be revised, such as the last page and the DOI of the references should be indicated consistently.

Response 9. Appropriate corrections have been made.

Reviewer 2 Report

This manuscript is describing about hemolytic activity and cytotoxicity of synthetic nanoclays with montmorillonite structure for medical applications. The authors investigated the effect of the aluminum content, particle sizes, porosity and  zeta-potential of the samples on their toxicity. Some of the results were interesting, however, most of the data showed negative results, i.e., increased the hemolytic activity and cytotoxicity by adding the Al. Therefore, it was no valuable for medical application and it is not acceptable for publication. 

- In abs, L14, studied -> was studied.

- Fig. 1 (d) 2 um scale should be changed to 1 um scale.

Author Response

Response to Reviewer 2 Comments

The authors would like to thank the Reviewer for reading the manuscript and for the comments and remarks made.

Point 1. The authors investigated the effect of the aluminum content, particle sizes, porosity and  zeta-potential of the samples on their toxicity. Some of the results were interesting, however, most of the data showed negative results, i.e., increased the hemolytic activity and cytotoxicity by adding the Al. Therefore, it was no valuable for medical application and it is not acceptable for publication.

Response 1. Natural montmorillonites have long been used in medicine, for example, as enterosorbents (e.g. “Smecta”, IPSEN CONSUMER HEALTHCARE, France) and components of wound dressings etc., despite their hemolytic activity and cytotoxicity. The advantage of synthetic montmorillonites lies in the ability to control their characteristics - chemical composition, particle size, porosity, surface properties, etc. The study of the effect of these characteristics on biological activity opens up wide opportunities for expanding the areas of application of such materials in medicine. In this work, it was shown that an increase in the aluminum content can lead to an increase in hemolytic activity and cytotoxicity. At the same time, significant toxicity appears at a sufficiently high concentration of samples, which will almost never be achieved in real conditions. Aluminum-free samples are generally non-toxic over a very wide range of concentrations. Thus, the results obtained make it possible to determine the compositions and concentrations that can be used to solve specific problems in medicine. The choice of a specific composition of synthetic montmorillonite to solve a specific problem will need to be made taking into account the results of this study, as well as taking into account other characteristics of the samples - their adsorption capacity for a particular drug, the ability to biodegradation, and so on.

Point 2. - In abs, L14, studied -> was studied.

Response 2. Thank you very much for your comment. The appropriate correction has been made.

Point 3. - Fig. 1 (d) 2 um scale should be changed to 1 um scale.

Response 3. Changes have been made.

Reviewer 3 Report

The manuscript by Golubeva et al. describes a toxicity assessment on synthetic clays with montmorillonite structure, materials that might be applied for extracorporeal blood detoxification. The application is certainly important, but requires careful evaluation of cytotoxicity and specifically hemolytic activity, therefore this research deserves consideration. The paper is well written and scientifically reliable, but a few aspects in my opinion need some amendments.

- I have not understood why the Authors used an endothelial cell line to verify clay cytotoxicity: as their application is extracorporeal, besides hemolytic activity, the evaluation of any effect on other blood cells such as monocytes would improve the significancy of this research;

- Line 38: I would suggest the Authors to increase the cited references about intercalation of molecules into natural clays, as many works have been published on this topic, for example the paper by Dongzhi et al, Applied Clay Science Volume 119, Pages 277 - 283, 2016, or the one by Baldassari et al, Applied Clay Science Volume 31, Issues 1–2, Pages 134-141, 2006;

- Lines 45-46: “In the recent past, the sepsis ended in 45 death in 80 % of cases.” needs a reference;

- In the paper the Authors used “absorption” several times; I guess they meant “adsorption”;

- Lines 74-75: “Raw MT studies have shown that it can cause cytotoxic effects at high concentrations after long-time exposure.” needs a reference;

- The SI symbol for liter is “L”, with the capital letter. Please amend;

- In the S2-Description of the instruments and methods used, in the "Measurement of benzene adsorption", I guess the residual pressure was 1.33*10-3 Pa, not Pa*s;

- In the S2-Description of the instruments and methods used, in "Hemolytic test" reference 7 is not pertinent, as no hemolytic test is described in that paper. The Authors must remove this self-citation, and in my opinion the test should be briefly described;

- In Figure 1 SEM image d is different from the others in terms of “zoom”. I would suggest to replace it with one similar to the other 3, if available. This would allow to evaluate any difference in layered structure;

- Lines 132-133: "The negative charge of the surface gradually increases as the degree of substitution of magnesium for aluminum increases". This sentence is confusing: I would suggest “the zeta potential gets more negative when the degree of substitution…”;

- Lines 135-136: any hypothesis to explain why the CEC reaches a maximum when 50% of Mg is replaced by Al?

- Data in Table 3 lack confidence intervals.

Author Response

Response to Reviewer 3 Comments

The authors would like to thank the Reviewer for a careful reading of the manuscript and for valuable comments and remarks, which the authors tried to take into account and make the appropriate changes and additions.

Point 1. I have not understood why the Authors used an endothelial cell line to verify clay cytotoxicity: as their application is extracorporeal, besides hemolytic activity, the evaluation of any effect on other blood cells such as monocytes would improve the significancy of this research;

Response 1. Layered silicates with montmorillonite can be used not only extracorporeally. Very promising areas of their potential application are enterosorption, as well as various medical sorbents that are in direct contact with endothelial cells, for example, components of wound and burn dressings. Therefore, information about the activity of these compounds in relation to endothelial cells is of great importance, allowing to determine the possibilities and prospects for the use of synthetic montmorillonites

Point 2. Line 38: I would suggest the Authors to increase the cited references about intercalation of molecules into natural clays, as many works have been published on this topic, for example the paper by Dongzhi et al, Applied Clay Science Volume 119, Pages 277 - 283, 2016, or the one by Baldassari et al, Applied Clay Science Volume 31, Issues 1–2, Pages 134-141, 2006;

Response 2. The authors took into account the reviewer's suggestion and expanded the references.

Point 3. Lines 45-46: “In the recent past, the sepsis ended in 45 death in 80 % of cases.” needs a reference.

Response 3. The relevant reference has been added.

Point 4. In the paper the Authors used “absorption” several times; I guess they meant “adsorption”.

Response 4. Thank you very much for this comment. The authors have made appropriate corrections to the text.

Point 5. Lines 74-75: “Raw MT studies have shown that it can cause cytotoxic effects at high concentrations after long-time exposure.” needs a reference;

Response 5. The reference has been added.

 Point 6. The SI symbol for liter is “L”, with the capital letter. Please amend;

Response 6. Thank you for your comment. Corrections have been made.

Point 7. In the S2-Description of the instruments and methods used, in the "Measurement of benzene adsorption", I guess the residual pressure was 1.33*10-3 Pa, not Pa*s

Response 7. Thank you once again for your comment. Corrections have been made.

Point 8. In the S2-Description of the instruments and methods used, in "Hemolytic test" reference 7 is not pertinent, as no hemolytic test is described in that paper. The Authors must remove this self-citation, and in my opinion the test should be briefly described;

Response 8. The authors apologize for the mistake. Indeed, a link was erroneously given to the authors' article, in which there is no hemolytic test description. At present, this mistake has been corrected and a reference is given to another article by the authors, in which a description of the technique is given. In addition, the authors added an extended description of the methodology to the Supporting information file.

Point 9. In Figure 1 SEM image d is different from the others in terms of “zoom”. I would suggest to replace it with one similar to the other 3, if available. This would allow to evaluate any difference in layered structure

Response 9. Thank you for this comment. Changes have been made to the scale of the figure.

Point 10. - Lines 132-133: "The negative charge of the surface gradually increases as the degree of substitution of magnesium for aluminum increases". This sentence is confusing: I would suggest “the zeta potential gets more negative when the degree of substitution…”;

Response 10. Changes have been made to the text of the manuscript in accordance with the suggestions of the reviewer.

Point 11. Lines 135-136: any hypothesis to explain why the CEC reaches a maximum when 50% of Mg is replaced by Al?

Respponse 11. The appearance of an extremum may be related to the transition of the dioctrahedral structure of the samples to trioctrahedral with the increasing degree of substitution of magnesium atoms for aluminum. This process is clearly visible on X-ray diffraction patterns (Figure S1) and is expressed in the disappearance of the peak at 2θ=60.8, d=1.48 Å, (060))(Al0, Al0.2, Al0.5 samples) and the appearance at 2θ=62.3, d=1.52 Å, (330) )(Al1.2, Al1.8, Al1.9 samples). This process is most likely accompanied by a change in the distribution of active sites on the surface of montmorillonite, which leads to the appearance of an extremum on the CEC.

Point 12. Data in Table 3 lack confidence intervals.

Response 12. Сonfidence intervals are added to the table.

Reviewer 4 Report

In the authors' current work, a preliminary evaluation of the haemolytic and cytotoxic properties of synthetic nanoclays with a montmorillonite structure has been carried out. The following issues need to be addressed before being published.

1. The form in which units are written should be kept in a uniform format throughout the text, e.g. in line 234 and on page S3 mg/ml should be mg/mL.

2. Short line connectives that should not be used for symbols expressing ranges of values.

3. The number of replicates of each assay, and the number of samples should be added to the figure captions and table captions.

4. Lack of information on the manufacturer of the material and cells.

5. The assay for haemolytic and cytotoxic properties, although references are provided, a brief description of the procedure as well as the calculation formulae is recommended.

6. Additional data processing methods are available and it is recommended that the necessary statistical analysis be performed.

7. A previous report showed the assay for cytotoxicity (IET Nanobiotechnol. 2018 Dec;12(8):1037-1041. doi: 10.1049/iet-nbt.2018.5079.) and it is suggested to be cited in the revised manuscript.

Author Response

Response to Reviewer 4 Comments

The authors would like to thank the Reviewer for a thorough reading of the manuscript and for valuable comments.

Point 1. The form in which units are written should be kept in a uniform format throughout the text, e.g. in line 234 and on page S3 mg/ml should be mg/mL.

Response 1. Appropriate adjustments have been made. Units of measurement are presented in a uniform format

Point 2. Short line connectives that should not be used for symbols expressing ranges of values.

Response 2. Thank you for your comment. Corresponding amendments have been made to the manuscript.

Point 3. The number of replicates of each assay, and the number of samples should be added to the figure captions and table captions.

Response 3. The authors added this information to the description of the experimental procedures, as well as to the Supplementary Information File.

Point 4. Lack of information on the manufacturer of the material and cells.

Response 4. Information added.

 Point 5. The assay for hemolytic and cytotoxic properties, although references are provided, a brief description of the procedure as well as the calculation formulae is recommended.

Response 5. The Supplementary information file provides an extended description of the methods used (S2-S4).

Point 6. Additional data processing methods are available and it is recommended that the necessary statistical analysis be performed.

Response 6. Thank you very much for this recommendation. The authors will take it into account and in the future will use extended approaches of statistical analysis when processing experimental data.

Point 7. A previous report showed the assay for cytotoxicity (IET Nanobiotechnol. 2018 Dec;12(8):1037-1041. doi: 10.1049/iet-nbt.2018.5079.) and it is suggested to be cited in the revised manuscript.

Response 7. Thanks a lot for the suggested link. The authors studied this report and cited it in their manuscript (Reference 41).

Round 2

Reviewer 1 Report

Although there were many mistakes in the first version of this manuscript, the authors addressed the reviewers' comments carefully and can be accepted by this journal.

Reviewer 2 Report

It is recommended to be acceptable.